# Synergistic assembly of human pre-spliceosomes across introns and exons

Joerg E Braun[1,2], Larry J Friedman[2], Jeff Gelles[2]*, Melissa J Moore[1]*

[1]RNA Therapeutics Institute, University of Massachusetts Medical School, Worcester, United States; [2]Department of Biochemistry, Brandeis University, Waltham, United States

**Abstract** Most human genes contain multiple introns, necessitating mechanisms to effectively define exons and ensure their proper connection by spliceosomes. Human spliceosome assembly involves both cross-intron and cross-exon interactions, but how these work together is unclear. We examined in human nuclear extracts dynamic interactions of single pre-mRNA molecules with individual fluorescently tagged spliceosomal subcomplexes to investigate how cross-intron and cross-exon processes jointly promote pre-spliceosome assembly. U1 subcomplex bound to the 5' splice site of an intron acts jointly with U1 bound to the 5' splice site of the next intron to dramatically increase the rate and efficiency by which U2 subcomplex is recruited to the branch site/3' splice site of the upstream intron. The flanking 5' splice sites have greater than additive effects implying distinct mechanisms facilitating U2 recruitment. This synergy of 5' splice sites across introns and exons is likely important in promoting correct and efficient splicing of multi-intron pre-mRNAs.

DOI: https://doi.org/10.7554/eLife.37751.001

*For correspondence:
gelles@brandeis.edu (JG);
melissa.moore@umassmed.edu
(MJM)

Competing interests: The authors declare that no competing interests exist.

## Introduction

Spliceosomes consist of the U1, U2, and U4/U6.U5 small nuclear ribonucleoproteins (snRNPs) and multiprotein Prp19-complex as major building blocks, plus many transiently interacting splicing factors (*Wahl et al., 2009*). This machinery recognizes and assembles stepwise at splice sites (SS) (U1 at 5'SS and U2 at the 3'SS/branch site) to form pre-spliceosomes, which are subsequently remodeled into catalytically active spliceosomes. Pre-spliceosomes can form on multi-intron pre-mRNAs through at least two different pathways. An intron can be recognized a) via cross-intron interactions leading directly to a catalytically active spliceosome, or b) via cross-exon interactions where the exons flanking an intron are first defined, after which cross-intron interactions between adjacent cross-exon complexes lead to spliceosome assembly (*Moldón and Query, 2010*). On human pre-mRNAs, which characteristically harbor multiple long introns and short exons, exon definition predominates (*Berget, 1995*; *Fox-Walsh et al., 2005*). Indeed, splicing is greatly enhanced when a 5'SS is present across the exon downstream of an intron, highlighting the importance of exon definition in humans (*Talerico and Berget, 1990*; *Yue and Akusjärvi, 1999*). Cross-exon pre-spliceosomes can transition into cross-intron pre-spliceosomes, each having a distinct protein composition; the latter can then productively splice the pre-mRNA (*Chiara and Reed, 1995*; *Schneider et al., 2010*). However, the mechanisms by which cross-intron and cross-exon pre-spliceosomes work together to facilitate pre-mRNA splicing remain unclear. In *S. cerevisiae*, where the cross-intron pathway predominates, single-molecule approaches have proven invaluable for elucidating the kinetic pathways and subcomplex dynamics involved in spliceosome assembly (*Semlow et al., 2016*; *Warnasooriya and Rueda, 2014*). Here we developed the tools necessary to implement colocalization single-molecule spectroscopy (CoSMoS) in human cell extracts and used this system to investigate the dynamic mechanism of cross-intron and cross-exon cooperation in human pre-spliceosome assembly.

**eLife digest** A gene is a segment of DNA that usually carries the information required to build a protein, the molecules responsible for most of life's processes. This DNA segment is organized in modules, with coding sections separated by portions of non-coding DNA known as introns.

When a gene is 'turned on', it gets faithfully copied into a molecule of pre-messenger RNA (pre-mRNA), which contains the alternating coding and non-coding modules. Before it can serve as a template to create a protein, this pre-mRNA must be processed and all the introns removed by a structure called the spliceosome. If this delicate process goes wrong, inaccurate protein templates are produced that may be damaging for the cell.

Spliceosomes are precise molecular 'scissors' that can recognize where a coding module stops and an intron starts, and then make a snip in the pre-mRNA to remove the non-coding sequence. The spliceosome is a complex molecular machine formed of numerous parts – including one known as U1 snRNP – that must come together. When a pre-mRNA has several introns, a spliceosome assembles anew for each of them.

Braun et al. designed a new method that allows them to 'tag' spliceosomes extracted from a human cell and follow them as they come together. The experiments show that spliceosomes working on different introns in the same pre-mRNA actually help each other out. As one assembles, this helps the spliceosome that processes the neighboring intron to get built. In particular, the U1 snRNPs processing nearby introns collaborate to promote the assembly and activity of the spliceosomes. This teamwork is likely important to guarantee that multiple introns are cut out quickly and accurately.

DOI: https://doi.org/10.7554/eLife.37751.002

## Results

### Single-molecule visualization of spliceosome assembly and function in human cell extract

We began by investigating whether human nuclear extracts can assemble catalytically-competent spliceosomes on surface-tethered pre-mRNA molecules. To do this, we utilized the pre-mRNA model substrate PIP85A (*Moore and Sharp, 1992*) (*Figure 1A*, *Table 1*). We refer to this RNA here as '*5i3*' to indicate that, reading in the 5'-to-3' direction, it contains a partial exon, a 5'SS ('*5*'), an intron (*i*), a 3'SS ('*3*'), and another partial exon. For this pre-mRNA 20 ± 2% (s.d.) was converted to spliced products after 40 mins in human cell line HEK293 nuclear extract (*Figure 1B*). No spliced products were observed in the absence of ATP, which is required for spliceosome assembly. To monitor splicing of individual *5i3* molecules, we incorporated a green-excited dye into the 5' exon, a red-excited dye into the intron, and biotin at the end of the 3' exon (*Figure 1C*). We sparsely deposited this pre-mRNA onto a streptavidin-functionalized glass surface, added nuclear extract and followed green and red fluorescence from single pre-mRNA molecules over time. To exclude pre-mRNAs that lost intron signal due to RNA degradation, we selected only those molecules retaining 5' exon (green) fluorescence at the end of the 40 min experiment. Of these, 1.1 ± 0.7% (s.e.) lost intron (red) fluorescence in a control conducted in the absence of ATP (likely due to photobleaching) whereas 18 ± 4% (s.e.) lost intron fluorescence in the presence of ATP. This single-molecule assay specifically monitors intron release, while the bulk assay measures the second splicing step. Intron release requires an ATP-dependent structural rearrangement of the splicing machinery after the second splicing step, so as expected the lag phase was somewhat longer in the single-molecule assay (*Figure 1C*) than in the bulk splicing assay (*Figure 1B*). However, surface-tethered *5i3* pre-mRNA molecules were spliced with a similar efficiency (17%) to what is observed in solution (20%).

To enable single-molecule visualization of individual spliceosomal subcomplexes, we generated three HEK293 cell lines, each stably expressing C-terminally fSNAP-tagged U1-70K, U2B', or Snu114 at a level comparable to the endogenous protein (*Figure 1—figure supplement 1*). Co-immunoprecipitation experiments confirmed efficient incorporation of the tagged protein into U1, U2, or U5 respectively (*Figure 1—figure supplement 2*). Treatment of nuclear extracts with a green-excited dye-benzylguanine conjugate resulted in highly specific labeling of the fSNAP-tagged proteins

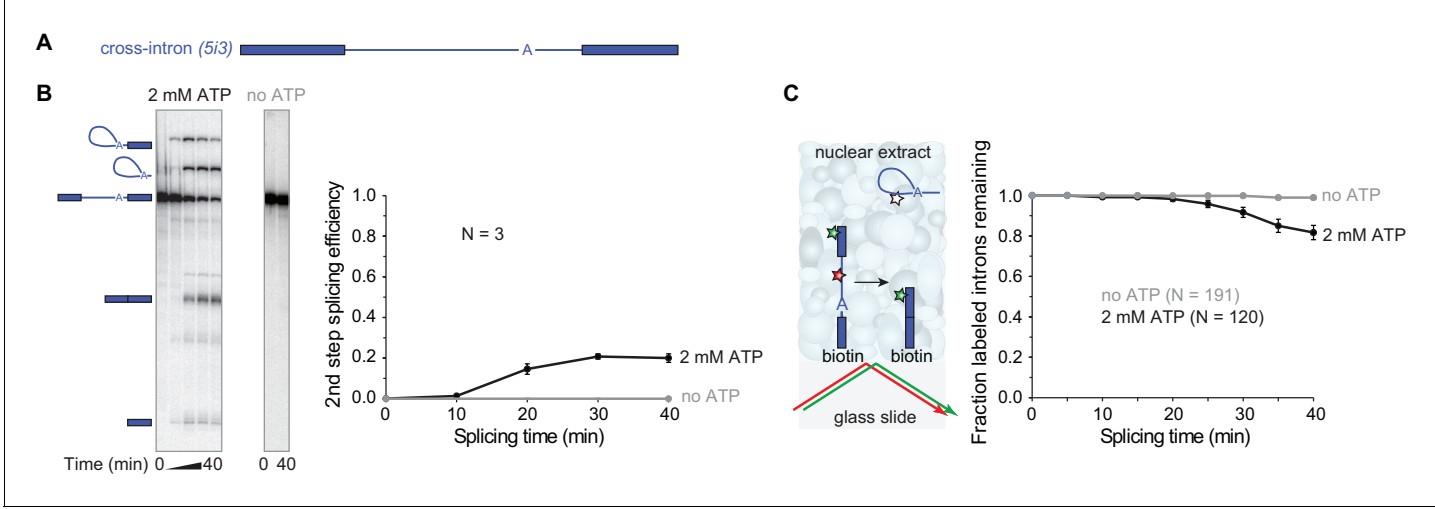

**Figure 1.** Observing splicing of the PIP85A (*5i3*) model pre-mRNA (**A**) in bulk (**B**) and single-molecule (**C**) splicing assays. Introns and exons are schematized as blue lines and rectangles, respectively, with A indicating the branchpoint. For bulk analysis (**B**) trace-labeled *5i3* was incubated with nuclear extracts, aliquots were analyzed on denaturing gel (15%) and second step splicing products quantified in graph. Second step splicing efficiency (±s.d.) was calculated as the amount of ligated exon product relative to the amount of *5i3* starting material at time zero. For single-molecule analysis (**C**), dyes tethered to the surface (red, green stars) were visualized using total internal reflection fluorescence microscopy using alternating red and green laser excitation (arrows); dye-labeled molecules in solution are not detectable. Fraction (±s.e.) of labeled introns remaining was calculated as the fraction of the N molecules retaining the exon dye fluorescence (green star) through the entire experiment duration which retained intron dye fluorescence (red star) at a particular time. Labeling of spliceosomal subcomplexes is shown in *Figure 1—figure supplements 1–5*.

DOI: https://doi.org/10.7554/eLife.37751.003

The following figure supplements are available for figure 1:

**Figure supplement 1.** Expression of fSNAP-fusion proteins in HEK293 cells.
DOI: https://doi.org/10.7554/eLife.37751.004

**Figure supplement 2.** Co-immunoprecipitation experiments show integration of fusion proteins into spliceosome assembly intermediates at efficiencies similar to their endogenous untagged counterparts.
DOI: https://doi.org/10.7554/eLife.37751.005

**Figure supplement 3.** Spliceosomal subcomplex labeling in human nuclear extracts.
DOI: https://doi.org/10.7554/eLife.37751.006

**Figure supplement 4.** Labeling efficiencies for spliceosomal proteins.
DOI: https://doi.org/10.7554/eLife.37751.007

**Figure supplement 5.** Bulk splicing efficiency and protein concentration for the nuclear extracts used in this study.
DOI: https://doi.org/10.7554/eLife.37751.008

(*Figure 1—figure supplement 3*). Because spliceosomal subcomplex concentrations in human nuclear extracts exceed the low dye concentrations optimal for single-molecule fluorescence, we labeled the tagged proteins using a limiting dye concentration (200 nM). Under these conditions, 30%, 60%, and 50% of total (tagged plus untagged) U1-70K, U2B', and Snu114 were labeled, respectively (*Figure 1—figure supplement 4*). Importantly, all tagged and dye-labeled extracts exhibited bulk splicing efficiencies comparable to extracts from the untagged parental cell line (*Figure 1—figure supplement 5*).

## Assembly of cross-intron and cross-exon pre-spliceosomes

We next used the labeled extracts in CoSMoS experiments in which we compared subcomplex dynamics on individual RNAs designed to promote assembly of cross-intron or cross-exon pre-spliceosomes (*Figure 2A*). The cross-intron RNA (*5i3*, *Figure 2B*) was identical to that in *Figure 1* except that it contained a single red-excited dye adjacent to the biotin tether at the end of the 3' exon. An identically-labeled cross-exon RNA (*3e5*, *Figure 2B*) was constructed by swapping the 5' and 3' halves of *5i3*. Thus, *5i3* and *3e5* consist of identical sequence segments, and differ only by whether the 5' and 3'SS are separated by an intron (*i*) or an exon (*e*). Simultaneous presence of the two RNA species at distinct, known locations on the slide surface allowed us to compare their

**Table 1.** RNAs.

5 N-U, 5-aminoallyluridine. U-DY547, Dylight 547 dye conjugated to C5 of U through a six-carbon linker. Small letters indicate mutated bases in the mutant sequence variants. The cryptic 5′SS in *Xi3e5* (see *Figure 4B*) is underlined.

| RNA name | RNA sequence (5′ to 3′) |
|---|---|
| *5i3* (cross-intron, PIP85A) | GGGCGAAUUCGAGCUCACUCUCUUCCGCAUCGCUGUCUGCG AGGUACCCUACCAGGUGAGUAUGGAUCCCUCUAAAAGCGGG CAUGACUUCUAGAGUAGUCCAGGGUUUCCGAGGGUUUCCG UCGACGAUGUCAGCUCGUCUCGAGGGCGUACUAACUGGGC CCCUUCUUCUUUUUCCCUCAGGUCCUACACAACAUACUGCA GGACAAACUCUUCGCGGUCUCUGCAUGCAA |
| *Xi3* (cross-intron, 5′SS mutant) | GGGCGAAUUCGAGCUCACUCUCUUCCGCAUCGCUGUCUGCG AGGUACCCUACCccccGAGUAUGGAUCCCUCUAAAAGCGGGCA UGACUUCUAGAGUAGUCCAGGGUUUCCGAGGGUUUCCGUCG ACGAUGUCAGCUCGUCUCGAGGGCGUACUAACUGGGCCCCUU CUUCUUUUUCCCUCAGGUCCUACACAACAUACUGCAGGACAA ACUCUUCGCGGUCUCUGCAUGCAA |
| *5iX* (cross-intron, polypyrimidine tract and 3′SS mutant) | GGGCGAAUUCGAGCUCACUCUCUUCCGCAUCGCUGUCUGCGA GGUACCCUACCAGGUGAGUAUGGAUCCCUCUAAAAGCGGGCAU GACUUCUAGAGUAGUCCAGGGUUUCCGAGGGUUUCCGUCGAC GAUGUCAGCUCGUCUCGAGGGCGUACUAACUGGGCCgCUaCaU gaUaUaCgCaCGGGUCCUACACAACAUACUGCAGGACAAACUCU UCGCGGUCUCUGCAUGCAA |
| *3e5* (cross-exon) | GGGCGAAUUCGUCGACGAUGUCAGCUCGUCUCGAGGGCGUAC UAACUGGGCCCCUUCUUCUUUUUCCCUCAGGUCCUACACAAC AUACUGCAGGACAAACUCUUCGCGGUCUCUGCAUGCGAGCUC ACUCUCUUCCGCAUCGCUGUCUGCGAGGUACCCUACCAGGU GAGUAUGGAUCCCUCUAAAAGCGGGCAUGACUUCUAGAGUAG UCCAGGGUUUCCGAGGGUUUCCAA |
| *3eX* (cross-exon, 5′SS mutant) | GGGCGAAUUCGUCGACGAUGUCAGCUCGUCUCGAGGGCGUA CUAACUGGGCCCCUUCUUCUUUUUCCCUCAGGUCCUACACA ACAUACUGCAGGACAAACUCUUCGCGGUCUCUGCAUGCGAG CUCACUCUCUUCCGCAUCGCUGUCUGCGAGGUACCCUACCc cccGAGUAUGGAUCCCUCUAAAAGCGGGCAUGACUUCUAGAG UAGUCCAGGGUUUCCGAGGGUUUCCAA |
| *Xe5* (cross-exon, polypyrimidine tract and 3′SS mutant) | GGGCGAAUUCGUCGACGAUGUCAGCUCGUCUCGAGGGCGU ACUAACUGGGCCgCUaCaUgaUaUaCgCaCgGGUCCUACACA ACAUACUGCAGGACAAACUCUUCGCGGUCUCUGCAUGCGA GCUCACUCUCUUCCGCAUCGCUGUCUGCGAGGUACCCUAC CAGGUGAGUAUGGAUCCCUCUAAAAGCGGGCAUGACUUCU AGAGUAGUCCAGGGUUUCCGAGGGUUUCCAA |
| *5i3e5* | GGGCGAAUUCGAGCUCACUCUCUUCCGCAUCGCUGUCUGC GAGGUACCCUACCAGGUGAGUAUGGAUCCCUCUAAAAGCGG GCAUGACUUCUAGAGUAGUCCAGGGUUUCCGAGGGUUUCC GUCGACGAUGUCAGCUCGUCUCGAGGGCGUACUAACUGGG CCCCUUCUUCUUUUUCCCUCAGGUCCUACACAACAUACUG CAGGACAAACUCUUCGCGGUCUCUGCAUGCGAGCUCACUC UCUUCCGCAUCGCUGUCUGCGAGGUACCCUACCAGGUGA GUAUGGAUCCCUCUAAAAGCGGGCAUGACUUCUAGAGUA GUCCAGGGUUUCCGAGGGUUUCCGACAAUUGCAUGAA |
| *5i3eX* | GGGCGAAUUCGAGCUCACUCUCUUCCGCAUCGCUGUCUG CGAGGUACCCUACCAGGUGAGUAUGGAUCCCUCUAAAAGC GGGCAUGACUUCUAGAGUAGUCCAGGGUUUCCGAGGGUU UCCGUCGACGAUGUCAGCUCGUCUCGAGGGCGUACUAAC UGGGCCCCUUCUUCUUUUUCCCUCAGGUCCUACACAACA UACUGCAGGACAAACUCUUCGCGGUCUCUGCAUGCGAGC UCACUCUCUUCCGCAUCGCUGUCUGCGAGGUACCCUACC ccccGAGUAUGGAUCCCUCUAAAAGCGGGCAUGACUUCUA GAGUAGUCCAGGGUUUCCGAGGGUUUCCGACAAUUGCAUGAA |

*Table 1 continued on next page*

*Table 1 continued*

| RNA name | RNA sequence (5' to 3') |
| --- | --- |
| Xi3e5 | GGGCGAAUUCGAGCUCACUCUCUUCCGCAUCGCUGUCU GCGAGGUACCCUACCccccGAGUAUGGAUCCCUCUAAAAG CGGGCAUGACUUCUAGAGUAGUCCAGGGUUUCCGAGGG UUUCCGUCGACGAUGUCAGCUCGCUCUCGAGGGCGUACU AACUGGGCCCCUUCUUCUUUUUCCCUCAGGUCCUACAC AACAUACUGCAGGACAAACUCUUCGCGGUCUCUGCAUG CGAGCUCACUCUCUUCCGCAUCGCUGUCUGCGAGGUAC CUACCAGGUGAGUAUGGAUCCCUCUAAAAGCGGGCAU GACUUCUAGAGUAGUCCAGGGUUUCCGAGGGUUUCCG ACAAUUGCAUGAA |
| Xi3eX | GGGCGAAUUCGAGCUCACUCUCUUCCGCAUCGCUGUC UGCGAGGUACCCUACCccccGAGUAUGGAUCCCUCUAA AAGCGGGCAUGACUUCUAGAGUAGUCCAGGGUUUCCG AGGGUUUCCGUCGACGAUGUCAGCUCGCUCUCGAGGGC GUACUAACUGGGCCCCUUCUUCUUUUUCCCUCAGGUC CUACACAACAUACUGCAGGACAAACUCUUCGCGGUCUC UGCAUGCGAGCUCACUCUCUUCCGCAUCGCUGUCUGC GAGGUACCCUACCccccGAGUAUGGAUCCCUCUAAAAGC GGGCAUGACUUCUAGAGUAGUCCAGGGUUUCCGAGGG UUUCCGACAAUUGCAUGAA |
| S1 | GGGCGAAUUCGAGCUCACUCUCUUCCGCAUCGCUGUCUG |
| S2 | CGAGGUACC(U-DY547)UACCAGGUGA |
| S3 | GUAUGGAUCCCUC(5 N-U)AAAAGCGGGCA(5 N-U)GACU UCUAGAG(5 N-U)AGUCCAGGGUUUCCGA |
| S4 | GGGUUUCCGUCGACGAUGUCAGCUCGCUCUCGAGGGCGU ACUAACUGGGCCCCUUCUUCUUUUUCCCUCAGGUCCUA CACAACAUACUGCAGGACAAACUCUUCGCGGUCUCUGCAUGCAA |

DOI: https://doi.org/10.7554/eLife.37751.027

behavior under identical experimental conditions within a single reaction chamber (*Figure 2C*). After introducing extract containing dye-labeled U1, we monitored the binding and dissociation of labeled U1 to individual RNA molecules and to control locations that had no RNA (e.g., *Figure 2D*). Similar experiments were performed using U2- and U5-labeled extracts.

To quantitatively characterize subcomplex binding to the surface tethered RNAs, we measured both the frequency of RNA-specific subcomplex binding events and the steady-state fractional occupancy RNA molecules by the subcomplex. U1 binding to *5i3* RNA was highly dynamic with many arrivals and departures per active RNA molecule (*Figure 2E–G*) and a high frequency of short (<50 s) binding events (*Figure 2F and H*). In comparison, progressively fewer *5i3* molecules bound U2 and U5 and the average occupancy of RNAs by these subcomplexes was also progressively lower (*Figure 2F and I*). Observed binding events were almost entirely RNA-specific; fewer and only short-duration binding events were observed at control 'no RNA' locations (*Figure 2—figure supplement 1*). Overall, the kinetics of human spliceosomal subcomplex binding and dissociation on the cross-intron *5i3* substrate were similar to those observed in CoSMoS experiments on cross-intron RNAs in *S. cerevisiae* extracts (*Hoskins et al., 2011*; *Shcherbakova et al., 2013*).

Spliceosomal subcomplex interactions with the cross-exon *3e5* RNA were quantitatively different from interactions with the cross-intron *5i3* RNA. *3e5* RNA molecules exhibited more frequent binding events (*Figure 2F and G*) and higher mean occupancy (*Figure 2I*) for all subcomplexes than *5i3* molecules. These differences might be partially explained by the inability of the cross-exon pre-mRNA to form a catalytically active spliceosome (*Figure 2—figure supplement 2*), resulting in the greater accumulation of inactive spliceosome assembly intermediates at steady state. Consistent with this idea, U1, U2 and U5 exhibited a higher frequency of long-lived (>50 s) binding events on the splicing-inactive *3e5* RNA than on the spliceable cross-intron *5i3* RNA (*Figure 2H*). In addition, possible differences between *5i3* and *3e5* three-dimensional structures could also alter the kinetics of their interactions with snRNPs.

We next investigated the effects of SS consensus sequence mutations on U1 and U2 binding dynamics. We restricted alterations to short consensus subcomplex binding sequences to reduce the

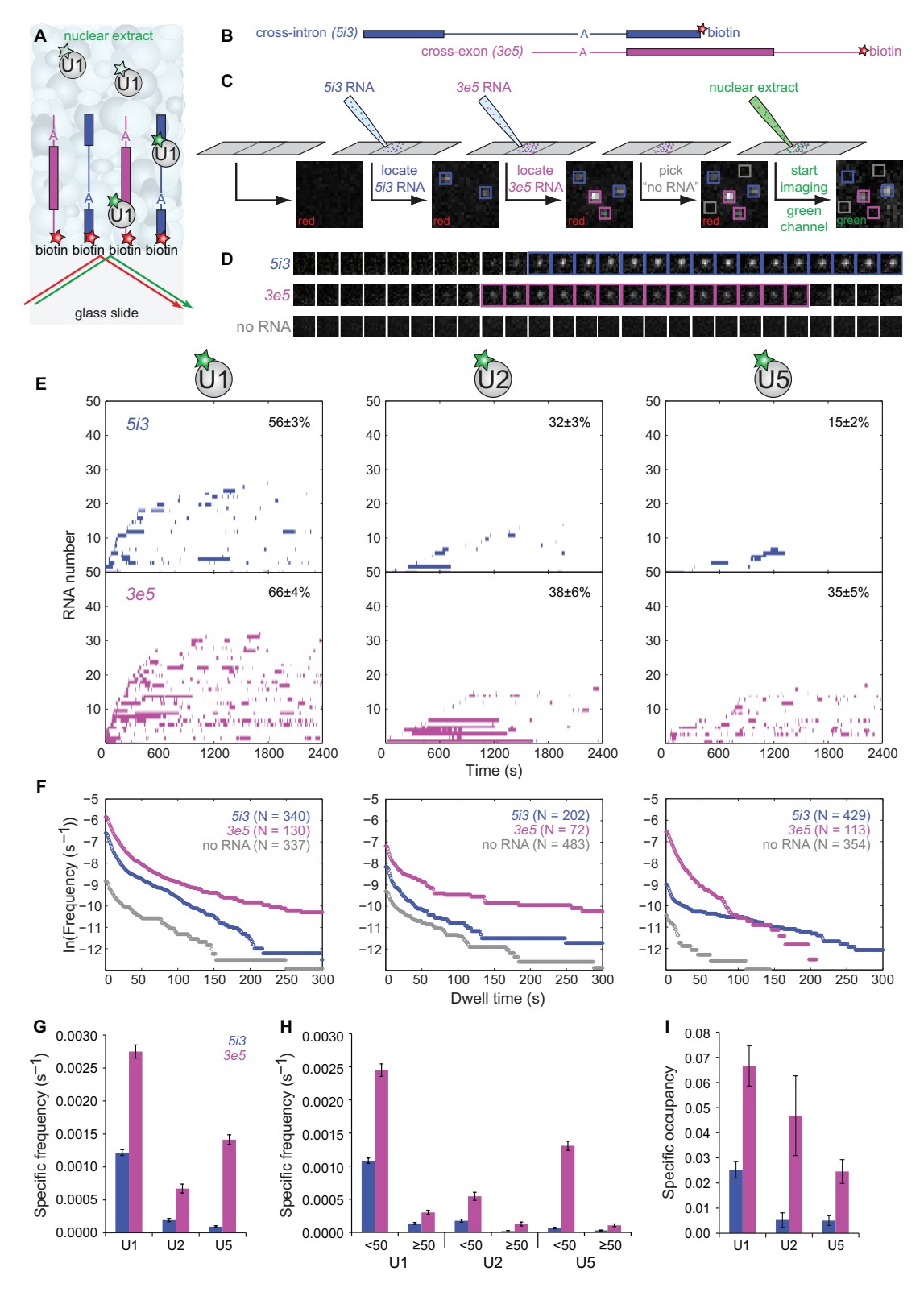

**Figure 2.** CoSMoS monitoring of spliceosomal subcomplex interactions with individual cross-intron (*5i3*) and cross-exon (*3e5*) pre-mRNA molecules in human nuclear extracts. (**A**) Schematic of a CoSMoS experiment in which green dye-labeled U1 is observed binding to red dye-labeled, surface-tethered RNAs. Introns and exons are schematized as blue and magenta lines and rectangles, respectively, with A indicating the branchpoint. Dyes (stars) linked to tethered RNAs were visualized using total internal reflection fluorescence microscopy using alternating red and green laser excitation

*Figure 2 continued on next page*

*Figure 2 continued*

(arrows); dye-labeled molecules in solution are not detectable. (B) Schematic of *5i3* and *3e5* RNAs, with features indicated as in (A). See *Table 1* for RNA sequences. (C) Protocol: *5i3* (blue) and *3e5* (magenta) RNAs were sequentially deposited and located (squares) under red laser excitation. Non-overlapping control 'no RNA' locations (gray) were selected. Then, extract was introduced and spliceosomal subcomplex (e.g., **U1**) binding to individual RNA molecules was visualized under green laser excitation. Images (grayscale) are a small portion (2.6 μm x 2.6 μm) of the microscope field of view recorded at each stage of the process. See *Figure 2—figure supplement 3* for complete field of view. (D) Time series images (1 s per frame; 1.3 μm x 1.3 μm) of U1 fluorescence from example surface locations containing a single *5i3* RNA (top), a single *3e5* RNA (middle) or no detected RNA (bottom). Images with fluorescence spots (highlighted) indicate U1 binding. See *Figure 2—figure supplement 4* for additional traces and detected events. (E) Rastergrams aggregating U1, U2, and U5 binding time courses from random samples of 50 individual *5i3* and *3e5* RNA molecules over 2,400 s. Each row in these plots contains data from a single RNA molecule; color indicates presence and white indicates absence of bound spliceosomal subcomplex. In each panel, RNA molecules are sorted by the time of first subcomplex binding (latest to earliest); the percentage (±s.e.) of N observed RNA molecules that exhibited subcomplex binding during the experiment is indicated. Rastergrams for 'no RNA' control locations are shown in *Figure 2—figure supplement 1*. (F) Cumulative distributions of U1, U2, and U5 dwell times on N observed *5i3* and *3e5* RNAs or control 'no RNA' locations. Data show the mean frequency per RNA molecule (or per 'no RNA' location) of subcomplex binding events with durations greater than or equal to the indicated dwell time. All frequencies on RNAs are substantially higher than the non-specific binding seen at 'no RNA' locations (note logarithmic scale). (G) Total frequencies (±s.e.) per RNA molecule of RNA-specific subcomplex binding. These RNA-specific binding frequencies correspond to the RNA minus the no RNA vertical axis intercepts of the curves in (F); they represent the total rate of subcomplex-RNA binding throughout the 2,400 s experiment averaged over all observed RNA molecules. (H) Frequencies (±s.e.) per RNA molecule of the subsets of RNA-specific subcomplex binding events shorter or longer than 50 s. (I) Specific occupancy (±s.e.), corresponding to the fraction of RNA molecules bound by the indicated fluorescent subcomplex averaged over the duration of the experiment. Numbers of RNA molecules observed in (G–I) are the same as those reported in (F). The specific occupancy values are calculated as described (see Materials and methods) to correct for the small amount of binding observed at 'no RNA' locations. Source data for *Figure 2*: SourceDataFigure2.zip.

DOI: https://doi.org/10.7554/eLife.37751.009

The following source data and figure supplements are available for figure 2:

**Source data 1.** Data from the single-molecule experiments.
DOI: https://doi.org/10.7554/eLife.37751.012
**Figure supplement 1.** Rastergrams of each subcomplex binding to 50 randomly selected 'no RNA' control locations.
DOI: https://doi.org/10.7554/eLife.37751.010
**Figure supplement 2.** Bulk assay detects no splicing of *3e5* pre-mRNA.
DOI: https://doi.org/10.7554/eLife.37751.011
**Figure supplement 3.** Complete microscope field of view (48 μm x 49 μm, grayscale) containing the region shown in *Figure 2C* (yellow box).
DOI: https://doi.org/10.7554/eLife.37751.013
**Figure supplement 4.** Sample fluorescence intensity traces and detected U1 binding events on *5i3* and *3e5* RNA molecules and at no RNA control locations (see *Figure 2*).
DOI: https://doi.org/10.7554/eLife.37751.014

possibility of the mutations causing large-scale changes in the three-dimensional structures of the RNAs. Functional U2 association with pre-mRNA depends on a polypyrimidine tract and the 3'SS AG (*Ruskin et al., 1988*). As expected, multiple pyrimidine to purine substitutions within the polypyrimidine tract combined with a 3'SS AG to GG mutation (*Figure 3A*) greatly decreased the frequencies of RNA-specific U2 binding events, reducing binding to near-background levels (*Figure 3—figure supplement 1*). U1 binding, however, was largely unaffected by these mutations, with U1 specific association frequencies and dwell time distributions on *5iX* and *Xe5* RNAs indistinguishable from those on *5i3* and *3e5*, respectively (*Figure 3B* and *Figure 3—figure supplement 2*). Thus, consistent with previous cross-intron data in yeast (*Séraphin and Rosbash, 1991*), human U1 binding is independent of U2 binding in both cross-intron and cross-exon contexts.

Functional U1 association is blocked by mutation of the 5'SS consensus from AG/GU to CCCC (*Roca et al., 2013*). As expected, this mutation (*Figure 3C*) decreased the U1 association rate and eliminated RNA-specific long-duration (>60 s) U1 binding events in both cross-intron and cross-exon contexts (*Figure 3—figure supplement 3*). Thus long-duration U1 binding events reflect its association with the canonical 5'SS, as reported previously in *S. cerevisiae* extract (*Larson and Hoskins, 2017*). In contrast, short duration U1 binding events are still present (although reduced in frequency) after 5'SS mutation and may reflect sequence non-specific interactions with the RNA.

Surprisingly, the 5'SS mutations affected U2 binding even more strongly than U1 binding. In both the cross-intron and cross-exon contexts, elimination of the canonical 5'SS decreased U2 binding frequency to background within experimental uncertainty (*Figure 3D* and *Figure 3—figure*

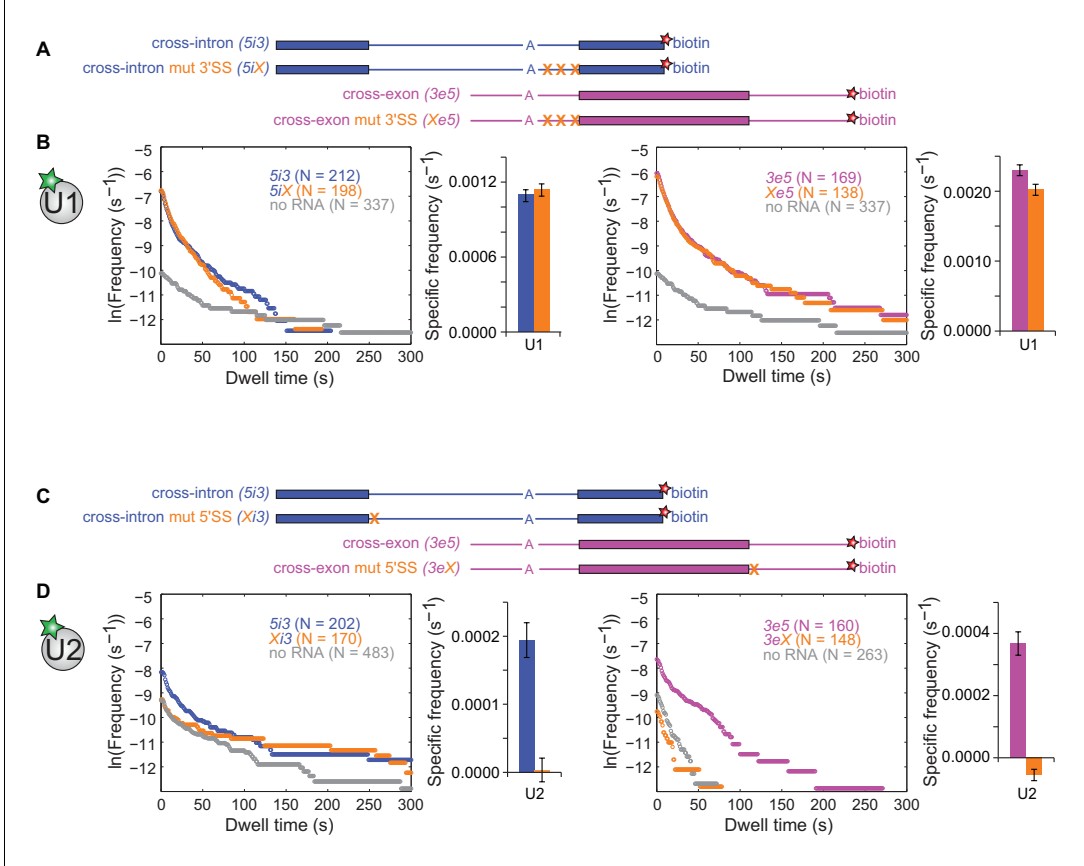

**Figure 3.** Interdependence of U1 and U2 binding to cross-intron and cross-exon RNAs. (**A**) Schematics of *5i3* and *3e5* RNAs without or with polypyrimidine tract and 3'SS mutations (**X**). See *Table 1* for RNA sequences. (**B**) Dynamics of dye labeled U1 binding to the RNAs depicted in (**A**), in a CoSMoS experiment in which all four RNAs were simultaneously present. Cumulative dwell time distributions and frequencies of RNA-specific binding were measured and plotted as in *Figure 2F and G*. For clarity, pairs of distributions are plotted in separate left and right panels and the no RNA data from the experiment is shown in both. Corresponding rastergrams are shown in *Figure 3—figure supplement 2*. (**C**) Schematics of *5i3* and *3e5* RNAs without or with 5'SS mutations (**X**). (**D**) Dynamics of dye-labeled U2 binding in two separate CoSMoS experiments, one with *5i3* and *Xi3* RNAs (left) and one with *3e5* and *3eX* (right). Corresponding rastergrams are shown in *Figure 3—figure supplement 4*. Dependence of U1 binding on 5'SS mutations and dependence of U2 binding on polypyrimidine tract and 3'SS mutations are shown in *Figure 3—figure supplement 1* and *Figure 3—figure supplement 3* respectively. Source data for the single-molecule experiments in *Figure 3*, *Figure 3—figure supplement 1*, *Figure 3—figure supplement 3*, *Figure 3—figure supplement 5*, and *Figure 3—figure supplement 6*: SourceDataFigure3.zip, SourceDataFigure3S1.zip, SourceDataFigure3S3.zip, SourceDataFigure3S5.zip, and SourceDataFigure3S6.zip.

DOI: https://doi.org/10.7554/eLife.37751.015

The following source data and figure supplements are available for figure 3:

**Source data 1.** Data from the single-molecule experiments.
DOI: https://doi.org/10.7554/eLife.37751.022
**Figure supplement 1.** U2 binding events to pre-mRNA depend on the polypyrimidine tract and 3'SS.
DOI: https://doi.org/10.7554/eLife.37751.016
**Figure supplement 1—source data 1.** Data from the single-molecule experiments.
DOI: https://doi.org/10.7554/eLife.37751.023
**Figure supplement 2.** Rastergrams showing U1 binding to 50 randomly selected *5i3*, *5iX*, *3e5*, and *Xe5* pre-mRNA molecules and 'no RNA' control locations over the course of 2,400 s, sorted by the time of the first binding event.
DOI: https://doi.org/10.7554/eLife.37751.017
**Figure supplement 3.** Long-duration U1 binding to pre-mRNA depends on a canonical 5'SS.
DOI: https://doi.org/10.7554/eLife.37751.018
**Figure supplement 3—source data 2.** Data from the single-molecule experiments.
DOI: https://doi.org/10.7554/eLife.37751.024
**Figure supplement 4.** Rastergrams showing U2 binding data on 50 randomly selected individual *5i3*, *Xi3*, *3e5*, and *3eX* pre-mRNA molecules and control 'no RNA' locations over the course of 2,400 s, sorted by the time of the first binding event to each RNA or location.

*Figure 3 continued on next page*

*Figure 3 continued*

DOI: https://doi.org/10.7554/eLife.37751.019

**Figure supplement 5.** U5 binding to *5i3* pre-mRNA depends on the 5'SS.

DOI: https://doi.org/10.7554/eLife.37751.020

**Figure supplement 5—source data 3.** Data from the single-molecule experiments.

DOI: https://doi.org/10.7554/eLife.37751.025

**Figure supplement 6.** Antisense morpholino oligonucleotide (AMO) targeting U1 snRNA strongly reduces U2 binding to both *5i3* and *3e5* RNAs.

DOI: https://doi.org/10.7554/eLife.37751.021

**Figure supplement 6—source data 4.** Data from the single-molecule experiments.

DOI: https://doi.org/10.7554/eLife.37751.026

*supplement 4*). Absence of U5 binding to these RNAs (*Figure 3—figure supplement 5*) confirmed that the mutations abolished the formation of functional pre-spliceosomes. These observations could indicate that observable U2 binding requires the U1 binding to a 5'SS positioned either cross-exon or cross-intron. Alternatively the results could also be explained if the 5'SS mutation indirectly affects U2 binding by affecting the pre-mRNA secondary structure. To exclude the latter possibility, we demonstrated U2 binding to the 5i3 and 3e5 was also eliminated in experiments (*Figure 3—figure supplement 6*) in which U1 interaction with the 5'SS was blocked by the addition of a morpholino oligonucleotide antisense to the U1 snRNA (*Kaida et al., 2010*). We conclude that U2 binding to the 3'SS is strongly dependent on U1 binding to a 5'SS either upstream (cross-intron) or downstream (cross-exon). This suggests an ordered human pre-spliceosome assembly pathway in which stable U2 association in the presence of ATP requires prior U1 binding, in contrast to the branched pathway observed in *S. cerevisiae* (*Shcherbakova et al., 2013*).

## Synergistic effects of cross-intron and cross-exon 5'SS on pre-spliceosome assembly

In multi-intron pre-mRNAs, internal exons have 5'SS both upstream and downstream, either or both of which could bind U1 and act to recruit U2 to the 3'SS. In principle, the two U1s could act either independently or synergistically. If there is only one means by which U1 can recruit U2 (e.g., via binding to a single site on U2), the combined effect of two U1s on U2 binding would be at most the sum of their individual actions (*Herschlag and Johnson, 1993*). Such is the case for the activities of multiple SR proteins on splicing efficiency (*Graveley et al., 1998*). In contrast, if the upstream and downstream U1s can interact with U2 simultaneously, or if they accelerate different steps in the overall U2 recruitment process, their combined effect could be larger, that is, synergistic. Such synergy was previously observed between two distinct sequence elements within a regulated splicing enhancer (*Lynch and Maniatis, 1995*).

To determine whether the upstream and downstream 5'SS independently or synergistically promote U2 binding, we constructed a pre-mRNA with 5'SS both upstream and downstream of the 3'SS (*5i3e5*, *Figure 4A*) as well as RNAs with mutations in either one 5'SS (*Xi3e5* and *5i3eX*) or both (*Xi3eX*). In these constructs, the sequences flanking both 5'SS were identical, minimizing potential sequence context effects. Consistent with the concept of exon definition (*Talerico and Berget, 1990*), splicing of *5i3e5* was 5-fold more efficient than *5i3eX* in ensemble splicing reactions (*Figure 4B*). In fact, the effect of adding the downstream 5'SS was strong enough to activate an otherwise dormant cryptic 5'SS in the *Xi3e5* construct. As expected, no splicing was observed on *Xi3eX* RNA.

We next performed single-molecule observations of U2 binding to these same four RNAs, all tethered and observed in a single reaction chamber to facilitate their direct comparison. Consistent with the observation that U2 binding in the cross-intron and cross-exon contexts depends on a 5'SS (*Figure 3C and D*), no U2 binding above background levels was observed for the RNA with no 5'SS (*Xi3eX*) (*Figure 4C* and *Figure 4—figure supplement 1*). Additionally, the shapes of the U2 dwell time distributions on *5i3eX* and *Xi3e5* (*Figure 4C*) were similar to those of the shorter *5i3* cross-intron and *3e5* cross-exon constructs (*Figure 2F*). Thus, the 5'SS dependence of U2 binding seen with the longer RNAs was qualitatively similar to that observed on the previously characterized shorter RNAs.

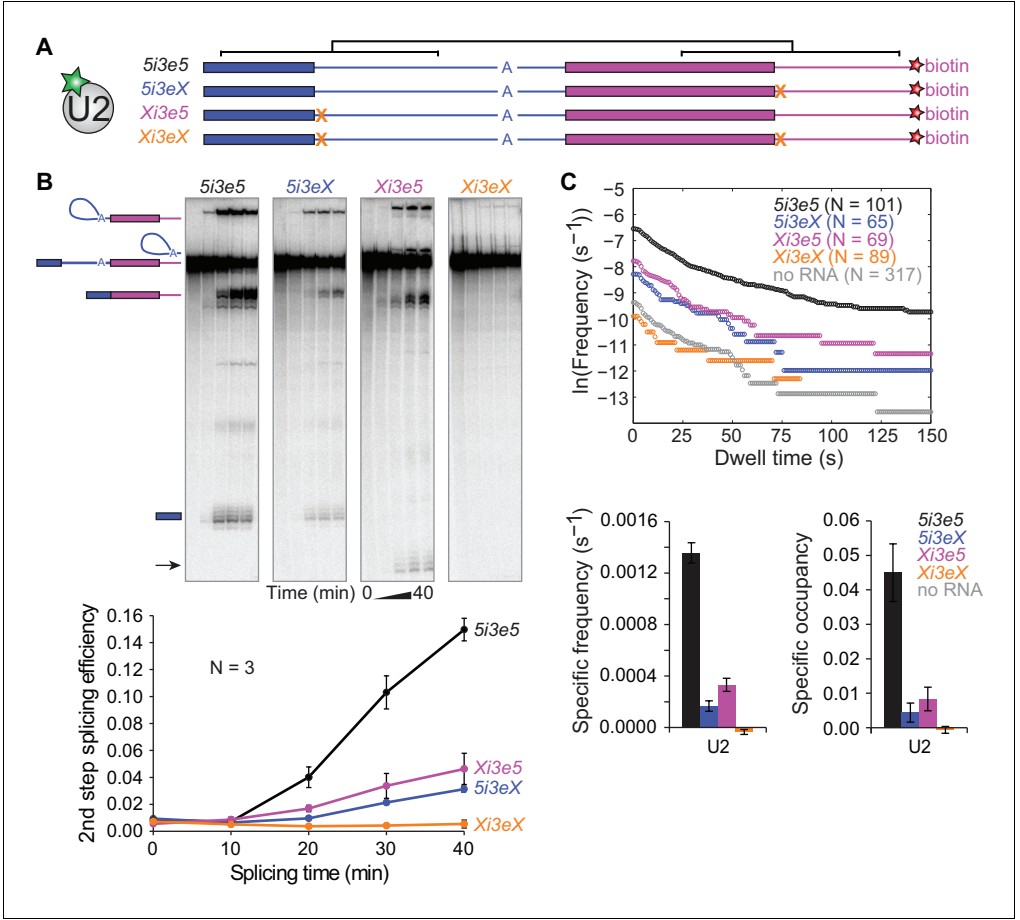

**Figure 4.** Synergistic recruitment of U2 by 5′SS across introns and exons. (A) Schematic of *5i3e5* RNAs without or with 5′SS mutations (X). Black brackets indicate two regions (113 nts) of identical sequence. (B) Ensemble splicing time courses of ³²P-labeled RNA. Second step splicing efficiencies (±s.d.) were calculated as fraction of *5i3e5*, *5i3eX*, *Xi3e5* and *Xi3eX* starting material at time zero. Arrow indicates 5′ exon resulting from usage of a cryptic 5′SS 12 nts upstream of the canonical 5′SS in *Xi3e5*. (C) Cumulative dwell time distributions, RNA-specific binding frequencies (±s.e.) and time-averaged fractional occupancies (±s.e.) of U2 binding to *5i3e5*, *5i3eX*, *Xi3e5* and *Xi3eX* RNAs measured in the same experiment. Corresponding rastergrams are shown in *Figure 4—figure supplement 1*. Analysis of time of first U2 binding event distributions is shown in *Figure 4—figure supplement 2*. Source data for the single-molecule experiments in *Figure 4*: SourceDataFigure4.zip.

DOI: https://doi.org/10.7554/eLife.37751.028

The following source data and figure supplements are available for figure 4:

**Source data 1.** Data from the single-molecule experiments.

DOI: https://doi.org/10.7554/eLife.37751.031

**Figure supplement 1.** Rastergrams showing U2 binding data on 50 randomly selected individual *5i3e5*, *5i3eX*, *Xi3e5*, and *Xi3eX* pre-mRNA molecules and control 'no RNA' locations from the experiment shown in *Figure 4C*, sorted by the time to the first binding observed on each RNA.

DOI: https://doi.org/10.7554/eLife.37751.029

**Figure supplement 2.** Cumulative distributions of the fraction of RNA molecules exhibiting at least one U2 binding event by the indicated time after the start of the experiment.

DOI: https://doi.org/10.7554/eLife.37751.030

Striking differences were apparent, however, when we compared the single 5′SS RNAs (*Xi3eX* and *5i3eX*) to the double 5′SS RNA (*5i3e5*) (*Figure 4C* and *Figure 4—figure supplement 1*). The presence of flanking 5′SS both upstream and downstream of the 3′SS dramatically increased U2 recruitment, with the RNA-specific U2 binding event frequency being more than 2.5 times the sum of the frequencies observed when either the upstream or downstream 5′SS was present alone. A

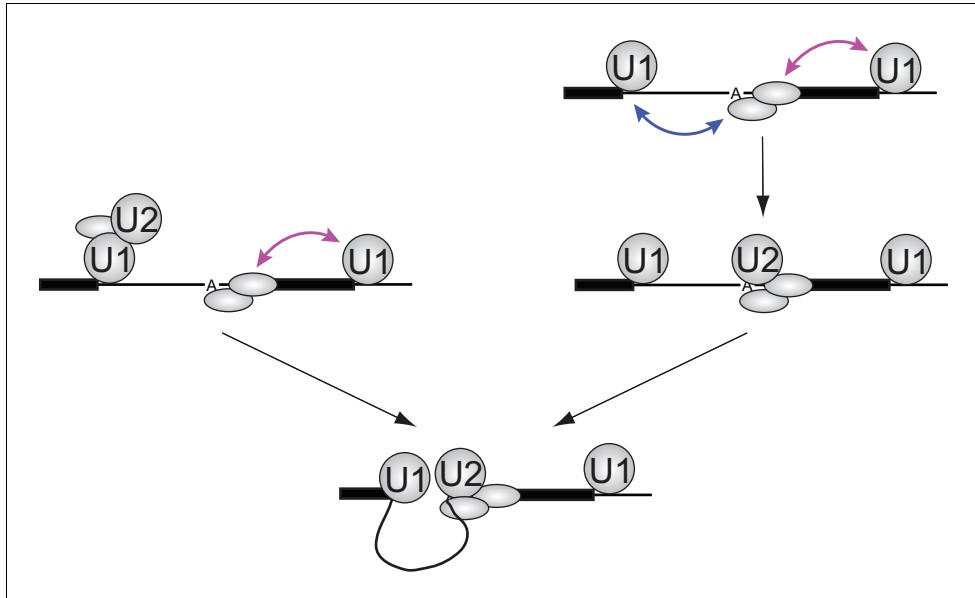

**Figure 5.** Implications of synergistic U2 recruitment for the mechanism of exon and intron recognition. The cartoon illustrates how differing modes of U1 action from upstream (cross-intron) and downstream (cross-exon) sites can synergize to promote faster U2 binding when both flanking U1 sites are present. Double-headed arrows denote physical interactions in which one component may accelerate association or slow dissociation of another.
DOI: https://doi.org/10.7554/eLife.37751.032

similar effect was seen in the U2 occupancy measurements (*Figure 4C*), although the occupancy data is more difficult to interpret due to splicing via the cryptic splice site in *Xi3e5*. A much greater than additive effect was also seen when we compared the distributions of the time to the first U2 binding observed on each RNA (*Figure 4—figure supplement 2*), a metric that is comparatively immune to artefacts from spot detection dropouts (*Friedman and Gelles, 2015*). Taken together, these data indicate that the upstream and downstream 5'SS act synergistically to accelerate the rate of stable U2 recruitment to the 3'SS.

## Discussion

Here we establish methods for observing the dynamics of spliceosomal subcomplexes on and the splicing of individual pre-mRNA molecules in human cell extract. The experiments reveal that the dynamics of U1 binding to the 5'SS are largely unaffected by the presence of U2 binding sites positioned either downstream (cross-exon) or upstream (cross-intron) of the 5'SS. In contrast, stable U2 binding to the 3'SS is accelerated by a 5'SS situated across either the adjacent intron or adjacent exon. Strikingly, when functional 5'SS are present together at both cross-intron and cross-exon locations, they synergistically promote U2 recruitment.

The more than additive effect of the flanking 5'SS indicates that U1 employs different molecular mechanisms/interactions across introns and across exons to accelerate U2 recruitment. A simple model that would explain this observed synergy of cross-exon and cross-intron 5'SS on U2 recruitment rate is that cross-exon U1 binding assists in recruiting the splicing factors that form a platform for U2 binding to the upstream 3'SS region, whereas the cross-intron U1-U2 interaction favors a U2 conformation capable of productive branch site engagement (*Figure 5*, left). Across exons, U1 is known to recruit U2AF (U2 auxiliary factor), which recognizes the polypyrimidine tract and 3'SS AG, and physically interacts with other proteins (e.g., SF1, p14, SF3B and SR proteins) required for stable U2 binding (*Black, 2003*). Across introns, U1 and U2 can interact via the DEAD-box protein Prp5, the ATPase activity of which promotes a structural change in U2 that makes the branch site recognition sequence more solvent accessible (*Abu Dayyeh et al., 2002*; *O'Day et al., 1996*; *Xu et al., 2004*). A different and not necessarily mutually exclusive model is that cross-exon and cross-intron

U1 interactions could both help form the binding platform for U2, but by interacting with different proteins (e.g., the upstream U1 stabilizes SF1 and the downstream U1 stabilizes U2AF) (*Black, 2003*). This arrangement would increase the likelihood of U2 encountering a fully assembled binding platform when in a conformation capable of stable branch site engagement (*Figure 5*, right). Other mechanisms for synergy, such as those mediated by effects on RNA secondary structure, are also possible.

Models of the type presented in *Figure 5* assume that once a functional U1-U2 pre-spliceosome forms across the intron, interactions with U1 bound to the downstream 5'SS provide no additional U2 binding stabilization or stimulation of subsequent steps of spliceosome assembly (*Figure 5*, bottom). Such models predict that the presence of a downstream U1 will not affect the lifetimes of stable U2 complexes once formed. The data in *Figure 4C* are consistent with this prediction: both the overall shape of dwell time distributions and the ratios of binding frequency to occupancy are similar for the RNAs with (*5i3e5*: $1.0 \pm 0.2$ s$^{-1}$) or without (*5i3eX*: $1.3 \pm 0.9$ s$^{-1}$) the downstream site. Taken

**Table 2.** Oligonucleotides and antisense morpholino oligonucleotides (AMO).
m, 2'-O-methyl ribonucleotide; 3ddN, 3' dideoxy nucleotide.

| Oligonucleotide name | Oligonucleotide sequence (5' to 3') |
|---|---|
| fSNAP-F | AGAGATAAGCTTTCCAGCGGTACCGAGCTCGGATC CAGCGGACCTAGGGAAACCTGCGGCCGCGGCTCCG GAGGCTCCGGCGGGAGCGGCATGGACAAAGACTGCGAAATG |
| fSNAP-R | ACAGATCTCGAGCTAACCCAGCCCAGGCTTGCCCAGTC |
| U1-70K-F | AGAGATGGTACCATGACCCAGTTCCTGCCGCCCAAC |
| U1-70K-R | ACAGATGCGGCCGCACTCCGGCGCAGCCTCCATC |
| U2B'-F | AGAGATGGTACCATGGATATCAGACCAAATCATAC |
| U2B'-R | ACAGATGCGGCCGCATTTCTTGGCGTATGTAATTTTC |
| Snu114-F | AGAGATGGTACCATGGATACCGACTTATATGATGAG |
| Snu114-R | ACAGATGCGGCCGCACATGGGGTAATTGAGCACAACATC |
| ligation splint | ACATCGTCGACGGAAACCCTCGGAAACCCTGGACTACT CTAGAAGTCATGCCCGCTTTTAGAGGGATCCATACTCA CCTGGTAAGGTACCTCGCAGACAGCGATGCGGAAGAG |
| S1-T7-F | TAATACGACTCACTATAGGGCGAATTCGAGCTCAC |
| S4-T7-F | TAATACGACTCACTATAGGGTTTCCGTCGACGATGTCAGCTC |
| S1-R | mCmAGACAGCGATGCGGAAG |
| S4-R | mUmUGCATGCAGAGACCGCGAAG |
| 2-color RNA Klenow splint | GTTCCTTGCATGCAGAGACCGCGAAGAG/3ddC/ |
| *5i3* template-F | TAATACGACTCACTATAGGGCGAATTCGAGCTCAC |
| *5i3* template-R | mUmUGCATGCAGAGACCGCGAAG |
| *5i3* Klenow splint | GTTCTTCTTATTGCATGCAGAGACCGCGAAGAG/3ddC/ |
| *5i3* Klenow capture | CTCTTCGCGGTCTCTGCATGCAATAAGAAGAAC |
| *3e5* template-F | TAATACGACTCACTATAGGGCGAATTCGTCGACG |
| *3e5* template-R | mUmUGGAAACCCTCGGAAACCCTG |
| *3e5* Klenow splint | GTTCTTCTTATTGGAAACCCTCGGAAACCCTGGA/3ddC/ |
| *3e5* Klenow capture | TCCAGGGTTTCCGAGGGTTTCCAATAAGAAGAAC |
| *5i3e5* template-F | TAATACGACTCACTATAGGGCGAATTC |
| *5i3e5* template-R | mUmUCATGCAATTGTCG |
| *5i3e5* Klenow splint | GTTCTTATCTTATTCATGCAATTGTCGGAAACCCTC/3ddC/ |
| *5i3e5* Klenow capture | GAGGGTTTCCGACAATTGCATGAATAAGATAAGAAC |
| control-AMO | CCTCTTACCTCAGTTACAATTTATA |
| anti-U1-AMO | GGTATCTCCCCTGCCAGGTAAGTAT |

DOI: https://doi.org/10.7554/eLife.37751.034

**Table 3.** Plasmids.

| Plasmid name | Description |
|---|---|
| pcDNA5-FRT-TetO | *Singh et al. (2012)* |
| pcDNA5-FRT-TetO-fSNAPc | open reading frame of fSNAP inserted into pcDNA5-FRT-TetO using the HindIII and XhoI restriction sites |
| pcDNA5-FRT-TetO-U1-70K-fSNAPc | open reading frame of U1-70K inserted into pcDNA5-FRT-TetO-fSNAPc using the KpnI and NotI restriction sites |
| pcDNA5-FRT-TetO-U2B''-fSNAPc | open reading frame of U2B'' inserted into pcDNA5-FRT-TetO-fSNAPc using the KpnI and NotI restriction sites |
| pcDNA5-FRT-TetO-Snu114-fSNAPc | open reading frame of Snu114 inserted into pcDNA5-FRT-TetO-fSNAPc using the KpnI and NotI restriction sites |
| PIP85.A (=T7-5i3) | *Moore and Sharp (1992)*, T7 transcription template for *5i3* |
| T7-3e5 | T7 transcription template for *3e5* |
| T7-5iX | T7 transcription template for *5iX* |
| T7-Xi3 | T7 transcription template for *Xi3* |
| T7-3eX | T7 transcription template for *3eX* |
| T7-Xe5 | T7 transcription template for *Xe5* |
| T7-5i3e5 | T7 transcription template for *5i3e5* |
| T7-5i3eX | T7 transcription template for *5i3eX* |
| T7-Xi3e5 | T7 transcription template for *Xi3e5* |
| T7-Xi3eX | T7 transcription template for *Xi3eX* |

DOI: https://doi.org/10.7554/eLife.37751.033

together, our data suggest that the synergistic stimulatory effect of the downstream 5'SS on splicing is exerted at the U2 recruitment step and the processes that enable it, not at the subsequent steps of spliceosome assembly and splicing.

Inappropriate skipping of otherwise constitutive internal exons can occur with exceptionally low frequency ($\sim$1 in $10^5$ splicing events) (*Fox-Walsh and Hertel, 2009*). But the molecular mechanisms contributing to this remarkable accuracy were previously unclear. By implementing CoSMoS in extracts from human cells expressing genetically-tagged proteins, we here show that a major contributor to exon inclusion is collaboration between flanking 5'SS. This collaboration dramatically increases the rate of stable U2 recruitment during pre-spliceosome formation. Cross-intron and cross-exon 5'SS synergy on U2 recruitment rate explains how U1 can have such a strong effect on U2 recruitment despite its association with pre-mRNA being much more dynamic than U2. This mechanism is likely crucial for rapid definition of internal exons in multi-intron RNAs, enabling the human splicing machinery to avoid inappropriate exon skipping.

# Materials and methods

## Nuclear extract preparation

Stable HEK293 Tet-On Flp-In cell lines were generated to express fSNAP fusions of U1-70K, U2B'', and Snu114 at near endogenous levels as previously described (*Singh et al., 2012*). Stable HEK293 Tet-On Flp-In cell lines were generated from the Flp-In T-REx−293 Cell Line (Invitrogen, R78007). The cells were purchased from Invitrogen and their resistance to Zeocin and their Flp-In competence were confirmed. No further authentication or mycoplasma contamination testing were performed. Plasmid pcDNA5-FRT-TetO-fSNAPc was generated by amplifying the open reading frame of fSNAP using PCR primers fSNAP-F and fSNAP-R (*Table 2*) and inserting into pcDNA5-FRT-TetO (Invitrogen) using the HindIII and XhoI restriction sites. Plasmids containing spliceosomal subcomplex protein-

fSNAP fusions (*Table 3*) were generated by PCR amplification from HEK293 cDNA for U1-70K and U2B'', and from Kazusa DNA Research Institute cDNA clone ORK00375 (*Nomura et al., 1994*) for Snu114 using the specified primers (*Table 2*) and cloning the products into pcDNA5-TetO-fSNAPc using the KpnI and NotI restriction sites. Expression levels of the fSNAP fusion proteins were adjusted to endogenous level by inducing the U1-70K-fSNAP, U2B''-fSNAP, and Snu114-fSNAP Flp-In cell lines with 6 ng/ml, 3 ng/ml, and 3 ng/ml Doxycycline (BD Biosciences, 631311), respectively. Parental cells were not induced. Nuclear extracts were prepared as previously described (*Lee et al., 1988*). In brief, HEK293 cells were grown at 37°C 5% $CO_2$ in DMEM medium supplemented with 10% FBS. Cells from 10 confluent 15 cm dishes were harvested and washed with ice cold PBS. Cells (900 µl) were resuspended in 900 µl Buffer A [10 mM Tris, 1.5 mM $MgCl_2$, 10 mM KCl and 0.5 mM DTT, pH 7.9 at 4°C, supplemented with complete protease inhibitor cocktail (Roche, 04693159001)], transferred to a 2 ml Eppendorf tube, incubated for on ice 15 min and then disrupted by 10 passages through a 25 gauge needle. After centrifuging the lysate for 20 s at $12,000 \times g$, the nuclear pellet was resuspended in 450 µl Buffer C [20 mM Tris, 25% (v/v) glycerol, 0.42 M NaCl, 1.5 mM $MgCl_2$, 0.2 mM EDTA and 0.5 mM DTT, pH 7.9 at 4°C, supplemented with complete protease inhibitor cocktail (Roche, 04693159001)] and rapidly stirred in a 2 ml round-bottom microcentrifuge tube with a $12.7 \times 3$ mm stir bar for 30 min. After clarifying the lysate by centrifuging for 10 min at $12,000 \times g$, the SNAP-Surface 549 dye-benzylguanine conjugate (New England BioLabs, S9112S) was added to the supernatant at a final concentration of 200 nM and incubated for 30 min at 30°C. After labeling, the supernatant was dialyzed 2 times for 2 hr each against Buffer E (20 mM Tris, 20% (v/v) glycerol, 0.1 M KCl, 0.2 mM EDTA and 0.5 mM DTT, pH 7.9 at 4°C). The dialysate (1 ml) was spun again for 10 min at $16,000 \times g$ and frozen in liquid nitrogen. Typical total protein concentration was 8.5 mg/ml. For negative control experiments, extracts were depleted of ATP using Centri-sep spin columns (Princeton Separations, CS-901) (*Anderson and Moore, 2000*). A single preparation of each of the three extracts was used in all reported experiments.

## RNA preparation

Radioactively labeled pre-mRNA substrate PIP85A (*Table 1*) was synthesized by in vitro transcription as previously described with a m7G(5')ppp(5')G 5' cap and [$\alpha$-$^{32}$P]UTP (*Moore and Sharp, 1992*).

Two color pre-mRNAs carrying a 3' biotin (*Figure 1C*) were prepared by splinted ligation as previously described (*Crawford et al., 2013*; *Shcherbakova et al., 2013*) using DNA oligonucleotides and RNA segments listed in *Tables 1* and *2*. Specifically, RNA segments S1 (with a m7G(5')ppp(5')G 5' cap) and S4 were produced in vitro by transcription by T7 RNA polymerase of templates generated by PCR from the plasmid PIP85A using the PCR primers S1-T7-F and S1-R or S4-T7-F and S4-R respectively. S2 and S3 were purchased from Dharmacon. Prior to the ligation 5' ends of S2, S3 and S4 were phosphorylated and S3 was labeled with AlexaFluor 647 NHS ester (Thermo Scientific, A20006) as previously described (*Crawford et al., 2013*; *Shcherbakova et al., 2013*). For the final splinted ligation S1, S2, S3, S4 RNA segments and the ligation splint oligonucleotide were annealed. The ligation resulted in an RNA where the 5' exon was labeled with a single green-excited dye (DY547) at position −7 relative to the 5'SS, the intron was labeled with on average two red-excited dyes (AlexaFluor 647) at positions 18, 30 and/or 42 relative to the 5'SS, and a single biotin was added to the 3' end by Klenow extension with biotin-dCTP (Trilink Bio Technologies Inc, N5002) (*Braun and Serebrov, 2017*; *Shcherbakova et al., 2013*).

One-color pre-mRNAs (*Figures 2*, *3* and *4*) were labeled at the 3' end by Klenow extension with both AlexaFluor 647 dUTP (Life Technologies, A32763) and biotin dCTP, resulting in one (*5i3* and *3e5*) or two (*5i3e5*) AlexaFluor 647 dyes and one biotin per pre-mRNA molecule. All oligonucleotides used for Klenow extensions are listed in *Tables 1* and *2*.

## Western blotting

Protein samples were separated by SDS-PAGE and transferred to a 0.45 µm pore size nitrocellulose membrane (Whatman, PROTRAN BA 85, 10 401 196). Proteins were detected using the indicated antibodies and an Odyssey CLx Imager (LI-COR) according to manufacturer's instructions.

## Co-immunoprecipitation

Cells (one 15 cm dish per condition) were lysed in 3 ml Buffer 1 (10 mM Tris pH 7.4, 100 mM NaCl, 2.5 mM MgCl$_2$) supplemented with 40 µg/ml digitonin. Nuclei were collected by pelleting at 2,000 × $g$ for 10 min and resuspended in 3 ml Buffer 1 supplemented with 0.1% Triton X-100 and complete protease inhibitor cocktail (Roche, 04693159001). The suspension was sonicated (Branson Digital Sonifier-250) for 8 s in bursts of 2 s and the NaCl concentration adjusted to 150 mM. This nuclear lysate was clarified by centrifugation at 15,000 × $g$ for 10 min and an input sample taken. Dynabeads Protein A (Life Technologies, 10002D) or Protein G (Life Technologies, 10001D) pre-incubated with respective antibodies were added and nutated for 2 hr. After four washes with Buffer 2 (20 mM Tris pH 7.4, 150 mM NaCl, 0.1% NP-40), bound proteins were eluted with SDS loading dye and analyzed by Western blotting as described above.

## Fluorescence and Coomassie gels

Protein samples were separated by denaturing polyacrylamide gel electrophoresis (SDS-PAGE). Gels were fixed in 25% isopropanol and 10% acetic acid, and fluorescence was imaged using a Typhoon scanner (GE Healthcare). Gels were subsequently stained with Coomassie Brilliant Blue dye R-250 (Thermo Scientific, 20278) to visualize total protein.

## Bulk in vitro splicing assays

Splicing reactions were performed at 30°C in 20 µl of 40% HEK 293 nuclear extract in final concentrations of 60 mM K$^+$-MOPS pH 7.3, 2 mM ATP, 0.5 mM DTT, 2 mM MgOAc$_2$, 20 mM potassium glutamate, 5 mM creatine phosphate, and 0.1 mg/ml *E.coli* tRNA with 20 fmol radioactively labeled pre-mRNA substrate. To make conditions correspond to those in the CoSMoS experiments, the bulk assays also included 0.9 U/ml *B. cepacia* protocatechuate dioxygenase (Sigma P8279; 5 U/mg; 9 mg/ml) and 5 mM protocatechuate (Sigma 37580, recrystallized from hot water before use) as an O$_2$ scavenging system and 1 mM Trolox (6-hydroxy-2,5,7,8-tetramethylchroman-2-carboxylic acid, Aldrich, 23,881–3) as a triplet quencher (*Hoskins et al., 2011*). Where indicated, anti-U1 AMO or control AMO (Gene Tools, sequences are described in *Table 2*) were added at 10 µM final concentration as previously described (*Kaida et al., 2010*) and the splicing reaction was 20 min pre-incubated at 30°C prior to the addition of pre-mRNA substrate. After incubating at 30°C for times indicated, splicing reactions were stopped by adding 10 volumes of Stop Buffer (100 mM Tris-Cl$^-$, 10 mM EDTA, 1% SDS, 150 mM NaCl, and 300 mM sodium acetate, pH 7.5). RNAs were extracted and separated by denaturing polyacrylamide (15%) gel electrophoresis. The dried gel was phosphorimaged with a Typhoon PhosphorImager and RNAs quantified using ImageQuant with signal intensities being normalized to their U content. Splicing efficiencies were calculated as the ratio of spliced RNA product (i.e., ligated exons) to pre-mRNA starting material at time zero.

## Single molecule in vitro splicing assays

Glass slides and cover slips were prepared as described previously (*Friedman et al., 2006*) except that PEGylation was only allowed to proceed for 3 hr at room temperature after which slides and coverslips were washed with 50 mM potassium phosphate buffer pH 7.4, dried with N$_2$ gas and stored at −80°C until use. After assembly of reaction chambers with vacuum grease (up to five lanes per slide with a volume of ~25 µl each), individual lanes were rehydrated immediately before use with 50 mM potassium phosphate buffer pH 7.4.

Single-molecule fluorescence imaging used a micro-mirror total internal reflection fluorescence (TIRF) microscope with automatic focus (*Friedman et al., 2006*; *Hoskins et al., 2011*). Sample temperature was maintained at 30°C using a custom-built temperature control system (*Paramanathan et al., 2014*). Streptavidin-conjugated fluorescent beads (Life Technologies, T10711) were tethered to the surface (multiple beads per field of view) and were used as reference for stage drift correction. RNAs were tethered on the slide surface at a total density of ~0.2–0.5 fluorescent spots per µm$^2$. When multiple different RNA species were tethered sequentially, a microscope image was taken after each round of deposition to individually identify the molecules of each RNA species. Splicing reactions (60 µl) were assembled as described for the bulk assays above (but without pre-mRNA) and introduced into individual slide lanes by capillary action and wicking; imaging was initiated immediately after reaction loading. For experiments with the two color pre-mRNA we

acquired a 1 s duration frame every 5 min with 150 µW 633 nm (red) excitation except at the beginning and end of the experiment when we acquired one frame per second with 300 µW 532 nm (green) excitation. For experiments with the one color pre-mRNAs and labeled nuclear extracts we acquired sequences of 100 one-second duration frames with 300 µW 532 nm excitation alternating with a single one-second frame with 150 µW 633 nm excitation. All excitation powers are measured incident to the input micro-mirror.

## Single molecule data analysis

Data analysis was performed using custom software (https://github.com/gelles-brandeis/CoSMoS_Analysis; copy archived at https://github.com/elifesciences-publications/CoSMoS_Analysis) implemented in MATLAB (MathWorks) as previously described (*Friedman and Gelles, 2015*; *Hoskins et al., 2011*); locations of fluorescent spots were identified by image analysis using the spot-picker algorithm (*Friedman and Gelles, 2015*). Locations of tethered RNA molecules and control locations were determined in drift-corrected, color-aligned images. For spliceosomal subcomplex detection, images were averaged with a five frame sliding window before spot picking. Binding frequencies were calculated as described (*Friedman and Gelles, 2015*). The provided source data files are 'intervals' files readable by imscroll (https://github.com/gelles-brandeis/CoSMoS_Analysis). The time-averaged specific occupancy of RNA molecules by a spliceosomal subcomplex (i.e., the fraction of time the RNA a fluorescently labeled subcomplex is bound to the RNA) was calculated as $(f_m - f_c) / (1 - f_c)$ where the subscripts m and c refer to RNA and no RNA control locations, respectively, and $f$ represents the fraction of time that a fluorescent spot was present, averaged over all locations measured for each type. Note that this value underestimates the subcomplex occupancy since only a fraction of each subcomplex is labeled (*Figure 1—figure supplement 4*). Standard error of the fractional occupancy was determined by bootstrapping (2,000 random samples). Distributions of time to first binding event were fit to background-corrected single-exponential models as described (*Friedman and Gelles, 2015*).

## Acknowledgements

We thank Eric Anderson and Andrew Franck for MATLAB scripts and Aaron Hoskins, Charles Query, and current/former Moore laboratory members for helpful discussions and critical manuscript review.

## Additional information

### Funding

| Funder | Grant reference number | Author |
|---|---|---|
| National Institutes of Health | R01 GM053007 | Melissa J Moore |
| Human Frontier Science Program | LT000166/2013 | Joerg E Braun |
| European Molecular Biology Organization | ALTF 890-2012 | Joerg E Braun |
| Howard Hughes Medical Institute | | Melissa J Moore |
| National Institutes of Health | R01 GM081648 | Jeff Gelles |

The funders had no role in study design, data collection and interpretation, or the decision to submit the work for publication.

### Author contributions

Joerg E Braun, Conceptualization, Resources, Data curation, Software, Formal analysis, Supervision, Funding acquisition, Validation, Investigation, Visualization, Methodology, Writing—original draft, Project administration, Writing—review and editing; Larry J Friedman, Resources, Data curation, Software, Formal analysis, Validation, Visualization; Jeff Gelles, Conceptualization, Resources, Data

curation, Formal analysis, Supervision, Funding acquisition, Validation, Investigation, Visualization, Methodology, Writing—review and editing; Melissa J Moore, Conceptualization, Resources, Data curation, Software, Formal analysis, Supervision, Funding acquisition, Validation, Investigation, Visualization, Methodology, Writing—review and editing

### Author ORCIDs
Joerg E Braun ⬦ http://orcid.org/0000-0002-8309-6401
Larry J Friedman ⬦ http://orcid.org/0000-0003-4946-8731
Jeff Gelles ⬦ http://orcid.org/0000-0001-7910-3421

### Decision letter and Author response
Decision letter https://doi.org/10.7554/eLife.37751.043
Author response https://doi.org/10.7554/eLife.37751.044

## Additional files

### Supplementary files
• Transparent reporting form
DOI: https://doi.org/10.7554/eLife.37751.035

### Data availability
All data generated or analysed during this study are included in the manuscript and supporting files. Source data files have been provided.

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
