## [Decision Letter]

[Editors’ note: a previous version of this study was rejected after peer review. The authors submitted a revised manuscript, which was accepted for publication without substantive changes. The decision letter after the initial peer review is shown below.]

Thank you for submitting your work entitled "Synergistic assembly of human pre-spliceosomes across introns and exons" for consideration by *eLife*. Your article has been reviewed by three peer reviewers, and the evaluation has been overseen by a Reviewing Editor and a Senior Editor. The following individuals involved in review of your submission have agreed to reveal their identity: Klemens Hertel (Reviewer #4).

Our decision has been reached after consultation between the reviewers. Based on these discussions and the individual reviews below, we regret to inform you that your work will not be considered further for publication in *eLife*.

Each referee found the work to be potentially exciting but all raised significant concerns that must be addressed. *eLife.* We encourage you to resubmit if and when the concerns are thoroughly dealt with. Although any resubmission will be treated as a new paper, it will be reviewed by the same scientists that examined the original submission.

Reviewer #2:

This manuscript by Braun et al., describes an exciting and carefully executed set of experiments that provide direct evidence for a phenomena that has long been observed but difficult to quantitatively demonstrate. Specifically, this study uses the single molecule CoSMoS approach developed in the Moore and Gelles labs to investigate the role of cross-intron and cross-exon interactions in the binding of the U1 and U2 snRNP; an early step in the assembly of the spliceosome. Many previous studies have demonstrated that in mammalian systems, having binding sites for both U1 and U2 either across an intron or on either side of an exon promotes spliceosome assembly and, ultimately, splicing itself. This has been interpreted to indicate that U1 and U2 interact in a cross-intron or cross-exon manner to promote stable association of one or the other component. Previous studies with CoSMoS have called into question such cooperative assembly of U1 and U2 in yeast; however, here the authors address this question in a mammalian system.

The experiments are elegantly designed, in the use of consistent sequences in the various constructs and in the analysis of multiple substrate variants on the same slide to ensure all RNAs are experiencing the same conditions. Moreover, the major conclusion – that stable association of U2 is synergistically promoted by both cross-intron and cross-exon interactions with U1 – is well supported. One could ask for additional mutations outside the splice sites as controls (for instance in a *5i3e5* context), but it is unclear that this would really add much to the overall story.

One conspicuously absent piece of data is the binding of U1 to *3eX* or *Xi3eX*. At least the first of these is critical if the authors are going to conclude that "U2 binding was even more strongly affected by the 5’SS mutations than U1". It would also be interesting to know how mutation of a single 5’SS in the *5i3e5* context impacts dwell times and frequencies of U1 overall. Finally, the authors also should provide data for the association of U2 with *Xe5*. One might imagine that this mutation doesn't reduce U2 frequency much if it is recruited via interaction with U1. This would be an interesting observation.

Reviewer #3:

In this manuscript, Braun et al. use single molecule fluorescence microscopy to study the formation dynamics of spliceosomal subcomplex assembly from human cell extracts. The authors measure the binding and dissociation kinetics between U1, U2 and U5 snRNPs and RNAs modelling cross-intron or cross-exon assembly. They use a surface immobilized pre-mRNA PIP81A. The same labs have pioneered the use of yeast cell extracts to study spliceosomal assembly at the single molecule level. Based on their data the authors propose that flanking U1 binding sites synergistically increase the rate of U2 binding.

Overall, the manuscript is well written and the experiments are technically sound. However, some of the conclusions in the manuscript appear to be only weakly supported by the data. In addition, aspects of the RNA change between experiments, making it difficult to make solid conclusions. For example, it is not clear how non-specific binding to non-splice sites affects the observed binding patterns, or how potential changes in secondary/tertiary structures have not been formally ruled out. Based on these issues, I cannot recommend publication of this manuscript in *eLife* in its current form. The conclusions may very well hold, but additional experiments/clarifications are need-ed in order to strengthen the paper.

Strengths

- Single molecule splicing assays with mammalian cell extracts is an important step forward in the field, which has been limited to yeast cell extracts for about a decade.

- The ability to begin addressing the question of cross-intron/exon spliceosomal assembly is important.

- The approach to monitor multiple RNAs on the same slide and under the same conditions is a strength of the paper.

Weaknesses

- All of the experiments presented here use a no-RNA control. In my mind this is not the best control for these experiments. An RNA with scrambled, modified or re-moved 5' or 3' splice sites should be used as a control instead.

- It is clear that many of the "short" (<50s) U1 binding events are non-specific. How these events are affecting the observed assembly pathways is an important question that needs to be solved in this manuscript. For example, control experiments with immunodepleted extracts (U1, etc.) could address this.

- Although *5i3* and *3e5* consist of the same sequence elements the order of the elements could dramatically affect their secondary and tertiary structures. How do the structures of these RNA affect the observed results?

- An important issue with the *3e5* experiments, is that there is no way to demonstrate that the observed binding events lead to bona fide splicing. Could the binding to these RNAs be completely off pathway?

- In transcribed pre-mRNAs cross-intron and cross-exon spliceosome assembly will directly compete with each other, the observed higher stability of the cross-exon assembly would suggest that cross-intron assembly would not take place in many instances, potentially leading to aberrant splicing. How do the authors reconcile their results with this?

- Subsection “Assembly of cross-intron and cross-exon pre-spliceosomes”, last paragraph: A simpler explanation is that the secondary structure of the *Xi3* RNA prevents binding of the U2 sub-complex.

Reviewer #4:

This manuscript describes a set of carefully designed single molecule experiments aimed at understanding spliceosomal component recruitment to introns and exons. The authors analyze U1, U2, and U5 association/dissociation dynamics in the context of an intron-defined pre-mRNA substrate, which can splice, and in the context of an exon flanked by functional splice sites, which cannot splice. The results of U1, U2 and U5 dynamics indicate that binding U1 is independent of U2 binding or of additional U1 binding sites. Interestingly, the data also show that U2 dynamics are significantly influenced by additional U1 binding. It is shown that the non-spliceable construct *3e5* exhibited higher U1, U2, and U5 occupancy when compared to the splicing competent *5i3*. The second major observation is that a downstream 5’SS (*5i3e5*) enhances splicing efficiency and U2 occupancy. When carrying out mutational analyses of *5i3e5*, deleting either 5’SS alone or in combination, the authors observe more than additive effects on U2 occupancy. The authors conclude that the presence of a downstream 5' splice site synergistically promotes U2 recruitment.

The experiments presented examine in more detail previous observations and they provide novel insights. Several years ago, it was demonstrated that downstream 5' splice sites have a positive effect on in vitro splicing (Chiara and Reed, 1995; Yue and Akusjarvi, 1999). It was argued then that their presence improves splicing much like splicing enhancers do. The present work takes such observations a step further by demonstrating that U2 occupancy is much improved. The manuscript should include a discussion of these previous works.

The main figures supporting the notion of synergy is Figure 4. Permutations of *5i3e5* were tested, either mutating one of the two 5’SS or both splice sites. Of concern here is the fact that *Xi3e5* (the upstream intronic 5’SS deleted) results in the activation of a cryptic splice site. *Xi3e5* is therefore no longer a substrate without an intronic 5’SS, but rather one that has a weaker intronic 5’SS. It is unclear to what degree the data obtained for this substrate may cloud the conclusions. By design, it was expected that this construct should not splice at all. Rather, the mutation should have rendered the splicing competent cross intron construct into a splicing incompetent cross exon construct, similarly as illustrated in Figure 2. Based on consistency expectations, one would have expected that the *Xi3e5* mutation display similar U2 occupancy as the *3e5* construct in Figure 2 – relatively high or ~4-fold higher than what is observed. Using the authors own arguments for explaining the occupancy difference between the splicing competent and incompetent substrates *3e5* and *5i3* (Figure 2), the low U2 signal of *Xi3e5* could be caused by its undesired splicing activity. This discrepancy needs to be addressed as the entire argument of synergy rests in part on the *Xi3e5* occupancy data point.

---

## [Author Response]

[Editors’ note: the author responses to the first round of peer review follow.]

We thank all three reviewers for their thorough analysis of our manuscript, and their many insightful and helpful comments. In response, we have added a substantial amount of new data and made numerous changes to the text. A summary of the major data additions is as follows:

1) We added data for U2 binding to *3e5* and *Xe5* pre-mRNAs (now Figure 3—figure supplement 1D-F). These data confirm that U2 binding requires a polypyrimidine tract and 3'SS also in the cross-exon context.

2) We added data on U1 binding to *3e5* and *3eX* pre-mRNAs (now Figure 3—figure supplement 3D-F). These data confirm that long-duration U1 binding depends on a canonical 5’SS also in the cross-exon context.

3) We added data showing that U5 binding to *5i3* pre-mRNA depends on the 5’SS (now Figure 3—figure supplement 5). These data confirm that the observed U5 binding events to *5i3* are sequence specific.

4) We added data showing that an antisense morpholino oligonucleotide (AMO) targeting U1 snRNA strongly reduces U2 binding to both *5i3* and *3e5* RNAs (now Figure 3—figure supplement 6). These results strongly support the hypothesis that the 5’SS mutations suppress U2 binding act by disfavoring U1 binding to the 5’SS rather than by altering pre-mRNA secondary structure.

Our detailed responses to individual reviewers' verbatim comments are given below:

Reviewer #2:[…] The experiments are elegantly designed, in the use of consistent sequences in the various constructs and in the analysis of multiple substrate variants on the same slide to ensure all RNAs are experiencing the same conditions. Moreover, the major conclusion – that stable association of U2 is synergistically promoted by both cross-intron and cross-exon interactions with U1 – is well supported. One could ask for additional mutations outside the splice sites as controls (for instance in a 5i3e5 context), but it is unclear that this would really add much to the overall story.

We agree that such mutations (presumably intended as negative controls) would not add much to the story. It is well established that there are effects of RNA sequences outside of the splice site consensus sequences on pre-spliceosome assembly, but comprehensive understanding of these is lacking. It would be difficult to rationally design such mutations and to interpret their effects.

One conspicuously absent piece of data is the binding of U1 to 3eX or Xi3eX. At least the first of these is critical if the authors are going to conclude that "U2 binding was even more strongly affected by the 5’SS mutations than U1".

We have now added the requested data on U1 binding to *3eX* as Figure 3—figure supplement 3D-F. These data confirm that, just as was demonstrated for the cross-intron RNA (*5i3*), mutation of the 5' splice site in the cross-exon context (*3e5*) essentially eliminates the long U1 binding events. With these new data, the statement quoted by the reviewer is now supported – in both cross-intron and cross-exon constructs, 5'-SS mutation reduces primarily long U1 binding events but reduces U2 binding events of all durations.

It would also be interesting to know how mutation of a single 5’SS in the 5i3e5 context impacts dwell times and frequencies of U1 overall.

Author response image 1 shows new data demonstrating the effect of both single and double 5’SS mutations on U1 binding in the *5i3e5* context. As expected, mutation of both 5’SS's reduces the long-duration (>~75 s) binding events to background levels. Long binding event frequencies are similar on *5i3e5, Xi3e5*, and *5i3eX*. Thus, in contrast to what we observe for U2, there is no evidence of 5’SS synergy for U1 recruitment. Based on the data in the manuscript, we expected many of the short binding events to be 5’SS-independent, and *Xi3eX* had a high frequency of short binding events. Despite the fact that these data show the expected results, we feel that some caution is required in interpreting them as we cannot distinguish between the presences of one or two U1's bound to each RNA molecule. In summary, we are happy to show these data here, but because they don't lend themselves to any straightforward conclusions, we prefer to leave them out of the manuscript to avoid unnecessary distraction for the reader.

**Author response image 1. respfig1:** U1 binding to four RNA constructs, all measured in the same experiment, analogous to the U2 data in Figure 4.

Finally, the authors also should provide data for the association of U2 with Xe5. One might imagine that this mutation doesn't reduce U2 frequency much if it is recruited via interaction with U1. This would be an interesting observation.

We have now added the requested data on U2 binding to *Xe5* as Figure 3—figure supplement 1D-F. These data confirm that, just as was demonstrated for the cross-intron RNA (*5i3*), mutation of the 3' splice site in the cross-exon context (*3e5*) greatly reduces (by ~10-fold) the frequency of U2 binding events. In our view, these results and the analogous cross intron results do *not* exclude a role of U1 in U2 recruitment, but simply show that U2 is usually not bound for long enough to be detected in our experiments unless a 3'SS is present to stabilize its association with the RNA.

Reviewer #3:[…] Weaknesses- All of the experiments presented here use a no-RNA control. In my mind this is not the best control for these experiments. An RNA with scrambled, modified or re-moved 5' or 3' splice sites should be used as a control instead.

The reviewer is correct that all of our experiments include a no RNA control; this control is important to make sure that we are correctly measuring the frequencies of binding to RNA molecules (both sequence-specific and sequence non-specific binding) as opposed to binding to the slide surface. Nonetheless, as suggested by the reviewer, the original manuscript also contained controls using RNAs with mutated splice sites. In the revised manuscript, we have now added even more controls of this type. Below is a detailed summary of the mutated splice site controls for experiments observing U1, U2, and U5 snRNPs. This list includes both controls present in the original manuscript and new experiments added in the revised version:

U2: Control experiments with *5iX* (Figure 3—figure supplement 1) and *Xe5* (now added to Figure 3—figure supplement 1D-F) show that mutation of the 3'SS consensus sequences reduce U2 binding to background. This implies that none of the detected binding of U2 is sequence nonspecific. Rather, all observed U2 binding over the no RNA background is 3'SS-dependent. Similarly, experiments with *Xi3* and *3eX* (Figure 3C,D) and *Xi3eX* (Figure 4C) show that mutation of the 5’SS consensus sequence(s) reduce(s) U2 binding to background. This again shows that none of the detected binding of U2 is sequence non-specific, and further reveals that all observed U2 binding over background is 5’SS-dependent.

U5: We have added new controls (Figure 3—figure supplement 5) showing that mutation of the 5’SS reduces U5 binding to background. This demonstrates that there is no detectable sequence non-specific binding of U5. Rather, as for U2, all detected U5 binding events depend on a 5’SS (presumably as a consequence of U1 binding there).

U1: Control experiments with *Xi3* (Figure 3—figure supplement 3A-C), *3eX* (now added as Figure 3—figure supplement 3D-F), and *Xi3eX* (shown in Author response image 1 in response to a comment by reviewer #2) clearly show that long U1 binding events, but not short ones, depend on the canonical 5’SS. This is not surprising, since U1 has additional roles aside from of prespliceosome formation – it is known to interact with proteins that bind the RNA cap structure, with proteins that bind the branch point, and with RNA sequences that diverge from the 5’SS consensus. Presumably most or all of the observed short binding events reflect these other interactions, as our experiments show that these short binding events are largely 5’SS independent. In contrast, the long events require the 5’SS, presumably due to sequence-specific interactions.

- It is clear that many of the "short" (<50s) U1 binding events are non-specific. How these events are affecting the observed assembly pathways is an important question that needs to be solved in this manuscript. For example, control experiments with immunodepleted extracts (U1, etc.) could address this.

We agree that our data do not speak to the question of whether the short events are involved in pre-spliceosome assembly. The major point of the manuscript is to establish the synergy of cross-intron and cross-exon interactions in recruiting U2 to the pre-spliceosome. We respectfully suggest that it is not necessary to resolve the role of the short U1 binding events to reach that conclusion from the data. Nonetheless, we have revised the text to explicitly state that some or all of the short events may reflect sequence non-specific interactions of U1 with RNA. It is possible that these short U1 binding events are irrelevant to splicing. Alternatively, it might be that the short events report the formation of interactions (e.g., sequence-non-specific binding) that can assist with the ultimate formation of a stable interaction with the 5’SS. With regard to additional control experiments, it was unclear to us how immunodepletion experiments would resolve this point. Nevertheless, we have now added to the manuscript an experiment in which we prevented specific U1 binding to the 5’SS by preincubating the extract with an antisense morpholino oligo directed at the 5' end of U1 snRNA.

- Although 5i3 and 3e5 consist of the same sequence elements the order of the elements could dramatically affect their secondary and tertiary structures. How do the structures of these RNA affect the observed results?

Yes, *5i3* and *3e5* could have different 2D and 3D structures despite containing the same sequence elements. We now explicitly state this in the manuscript (subsection “Assembly of cross-intron and cross-exon pre-spliceosomes”). However, an analysis of *5i3* and *3e5* RNA sequences using the Vienna RNA Websuite (Gruber et al., 2008 NAR 36:W70-4) for minimum free energy predictions suggests that *5i3* and *3e5* have very similar stability and structure elements as shown below.

The optimal secondary structure for *5i3* has a minimum free energy of **-**69.0kcal/mol. This structure is shown in Author response image 2 in dot-bracket notation with the second half highlighted in gray.

**Author response image 2. respfig2:** Optimal secondary structure prediction for *5i3*

The optimal secondary structure for *3e5* has a minimum free energy of **-***67.4*kcal/mol. This structure is shown in Author response image 3 with the same sequence as in *5i3* highlighted in gray.

**Author response image 3. respfig3:** Optimal secondary structure prediction for *3e5*

To aid comparison between the two structures, in Author response image 4 we have rearranged the halves to place them next to one another. The first line of dot-bracket notation is for *5i3* and the second is for *3e5*.

**Author response image 4. respfig4:** Comparison of the optimal secondary structure predictions of *5i3* and *3e5*

This analysis shows that the base pairing and the overall stabilities of the most stable secondary structures of *5i3* and *3e5* are very similar. Of course, the possibility exists that these calculated lowest free energy structures do not accurately represent the structures assumed by these sequences in our experiments. Thus it is important to note that we base all essential conclusions of the paper on comparisons of a particular RNA construct *with the same construct containing a localized binding site mutation*. This greatly reduces (but, we acknowledge, does not entirely eliminate) the possibility that differences in 3D structure might contribute to some of the results. We have now added explicit discussion of this experimental design feature (i.e., use of binding site mutations) to the text (subsection “Assembly of cross-intron and cross-exon pre-spliceosomes”).

- An important issue with the 3e5 experiments, is that there is no way to demonstrate that the observed binding events lead to bona fide splicing. Could the binding to these RNAs be completely off pathway?

The reviewer is correct that *3e5* does not splice (as is expected); the data confirming this can be found in Figure 2—figure supplement 4. Nevertheless, mutation of *3e5* to *3eX* essentially eliminates U2 binding as does mutation of *Xi3e5* to *Xi3eX* (compare Figure 3D, right to Figure 4C) and the experiments in Figure 4 demonstrate that both the upstream and downstream 5’SSs in *5i3e5* contribute synergistically to U2 recruitment and to splicing. Based on these data we think it reasonable to conclude that binding of U1 to the cross-exon 5’SS is on pathway for splicing of this RNA.

- In transcribed pre-mRNAs cross-intron and cross-exon spliceosome assembly will directly compete with each other, the observed higher stability of the cross-exon assembly would suggest that cross-intron assembly would not take place in many instances, potentially leading to aberrant splicing. How do the authors reconcile their results with this?

With respect to the higher stability of the cross-exon assembly, we assume that the reviewer is referring to the data of Figure 2I. As explained in the manuscript, these data do not indicate that cross-exon complexes are more stable than cross-intron, but merely show a higher steady-state concentration of the complexes due to the fact that there is no easy kinetic pathway out of the *3e5* complexes because they cannot splice. Furthermore, we are unaware of any basis for the suggestion that there is direct competition between cross-intron and cross-exon spliceosome assembly under conditions where splicing actually occurs. Indeed, it is not even clear that cross-exon and cross-intron complexes utilize the same binding surfaces on U2. The synergy we observe suggests that when a downstream, cross-exon 5’SS is present (i.e., in *5i3e5* vs. *5i3eX*), it greatly promotes both U2 recruitment and eventual splicing of the upstream intron rather than causing competition.

- Subsection “Assembly of cross-intron and cross-exon pre-spliceosomes”, last paragraph: A simpler explanation is that the secondary structure of the Xi3 RNA prevents binding of the U2 sub-complex.

We have now added data to the manuscript reporting experiments where, rather than preventing U1 binding to the 5’SS by mutating the 5’SS consensus (which, as the reviewer points out, has the potential to alter the secondary structure of the pre-mRNA), we instead prevented U1 binding to the 5’SS by adding a morpholino oligo complementary to U1 snRNA (using the same reagent as Kaida et al., 2010 Nature 468:664-668). This manipulation suppresses U2 binding even more strongly than the 5’SS mutations on both 5i3 and 3e5 RNAs (these data now added as Figure 3—figure supplement 6). These results strongly suggest that the 5’SS mutations suppress U2 binding act by disfavoring U1 binding to the 5’SS rather than by altering pre-mRNA secondary structure.

Reviewer #4:[…] The experiments presented examine in more detail previous observations and they provide novel insights. Several years ago, it was demonstrated that downstream 5' splice sites have a positive effect on in vitro splicing (Chiara and Reed, 1995; Yue and Akusjarvi, 1999). It was argued then that their presence improves splicing much like splicing enhancers do. The present work takes such observations a step further by demonstrating that U2 occupancy is much improved. The manuscript should include a discussion of these previous works.

Thank you for pointing out this oversight. We now cite these earlier results in the first paragraph of the revised manuscript.

The main figures supporting the notion of synergy is Figure 4. Permutations of 5i3e5 were tested, either mutating one of the two 5’SS or both splice sites. Of concern here is the fact that Xi3e5 (the upstream intronic 5’SS deleted) results in the activation of a cryptic splice site. Xi3e5 is therefore no longer a substrate without an intronic 5’SS, but rather one that has a weaker intronic 5’SS. It is unclear to what degree the data obtained for this substrate may cloud the conclusions. […] Using the authors own arguments for explaining the occupancy difference between the splicing competent and incompetent substrates 3e5 and 5i3 (Figure 2), the low U2 signal of Xi3e5 could be caused by its undesired splicing activity. This discrepancy needs to be addressed as the entire argument of synergy rests in part on the Xi3e5 occupancy data point.

We agree that this point required clarification; we have revised the final paragraph of the Results section accordingly. Author response image 5 and Author response image 6 show graphs that aggregate the relevant data from Figures 2 and 4. With respect to the specific frequency (a measure of the apparent association rate constant of U2 with the different RNAs), *5i3* and *5i3eX* are similar, as expected. In contrast, *3e5* is ~2-fold lower than *Xi3e5*. Since the latter contains all of the sequence elements present in the former, the most likely explanation is that there may be some additional secondary structure present in the latter that modestly slows U2 binding. Because of the possibility of such structural alterations (as well as experiment-to-experiment variability), we specifically chose not to compare RNAs of different lengths to reach our conclusions about synergy. Instead we compare RNAs that are otherwise identical except for consensus binding site mutations, using only data collected at the same time in the same sample (that is, under precisely identical conditions). (This point is also discussed in response to comments of reviewer 3, above.) Note that because it measures the *association* rate, the frequency measurement in Figure 4C is not significantly affected by consumption of the complexes resulting from the slow cryptic splicing of *Xi3e5*. The same is true for the time to first binding measurements (Figure 4—figure supplement 2). For these reasons, we base our argument for synergy on these analyses, which specifically tell us about the rate of formation of detectable U2-RNA complexes. That is why we emphasize both in the abstract and in the concluding paragraph of the paper that our data demonstrate synergy in *U2 recruitment* rather than talking about synergy in, for example, stabilization of the pre-spliceosome subsequent to U2 binding.

In contrast to the frequency data, mechanistic interpretation of the specific occupancy data is more difficult because it is affected by multiple factors, including possibly by cryptic splicing as pointed out by the reviewer. We now say this explicitly in the final paragraph of the Results section.

**Author response image 5. respfig5:** Total frequencies ( ± s.e.) per RNA molecule of RNA-specific U2 snRNP binding. Data are from Figures 2G and 4C.

**Author response image 6. respfig6:** Fractional occupancy ( ± s.e.) of RNA molecules by U2 snRNP, averaged over the duration of the experiment. Data are from Figures 2I and 4C.